# Effects of Ultra-High Pressure on Endogenous Enzyme Activities, Protein Properties, and Quality Characteristics of Shrimp (*Litopenaeus vannamei*) during Iced Storage

**DOI:** 10.3390/molecules27196302

**Published:** 2022-09-24

**Authors:** Chen Zhu, Dexin Jiao, Ying Sun, Lihang Chen, Siyu Meng, Xiaona Yu, Mingzhu Zheng, Meihong Liu, Jingsheng Liu, Huimin Liu

**Affiliations:** 1College of Food Science and Engineering, Jilin Agricultural University, Changchun 130118, China; 2National Engineering Research Center for Wheat and Corn Deep Processing, Jilin Agricultural University, Changchun 130118, China

**Keywords:** *Litopenaeus vannamei*, ultra-high pressure, endogenous enzymes, protein properties, quality characteristics

## Abstract

The present study aimed to explore the effects of ultra-high pressure (UHP) on the cathepsin (B, D, H, and L) activities, protein oxidation, and degradation properties as well as quality characteristics of iced shrimp (*Litopenaeus vannamei*). Fresh shrimps were vacuum-packed, treated with UHP (100–500 MPa for 5 min), and stored at 0 °C for 15 days. The results showed that the L* (luminance), b* (yellowness), *W* (whiteness), Δ*E* (color difference), hardness, shear force, gumminess, chewiness, and resilience of shrimp were significantly improved by UHP treatment. Moreover, the contents of surface hydrophobicity, myofibril fragmentation index (MFI), trichloroacetic acid (TCA)-soluble peptides, carbonyl, dityrosine, and free sulfhydryl of myofibrillar protein (MP) were significantly promoted by UHP treatment. In addition, UHP (above 300 MPa) treatment enhanced the mitochondrial membrane permeability but inhibited the lysosomal membrane stability, and the cathepsin (B, D, H, and L) activities. UHP treatment notably inhibited the activities of cathepsins, delayed protein oxidation and degradation, as well as texture softening of shrimp during storage. Generally, UHP treatment at 300 MPa for 5 min effectively delayed the protein and quality deterioration caused by endogenous enzymes and prolonged the shelf life of shrimp by 8 days.

## 1. Introduction

*Litopenaeus vannamei* is one of the vital economic prawns in the world, mainly produced in Asia and Latin America [1]. Previous studies have shown that *L. vannamei* is rich in water and protein, which provides favorable conditions for microbial growth [2]. After death of shrimp, microbiological spoilage and endogenous enzymatic such as cathepsin action accelerate deterioration of muscle quality, resulting in a rapid decline in freshness [3]. Frozen storage (between 3 and 6 months) is one of the most common ways to extend the shelf life of shrimp products. However, denaturation of proteins caused by freezing and thawing can directly damage the quality of the products, such as increased drip loss and texture deterioration [4]. Therefore, there is need for adoption of emerging methods such as use of the ultra-high pressure (UHP) technology to effectively meet the demands of consumers.

Previous studies have found that raw shrimp can turn black because of the action of polyphenol oxidase during long-term storage. In addition, heating shrimp to 100 °C also causes protein denaturation and cooking loss [5,6]. UHP is a non-thermal processing technology which inactivates live endogenous enzymes and microorganisms in aquatic products under a pressure between 100 and 1000 MPa. UHP treatment effectively delays the spoilage of aquatic products and ultimately prolongs their shelf life [7]. Moreover, UHP maintains the quality and storage stability of food by changing its composition and physicochemical properties [8].

Currently, several studies have shown the effects of UHP on the basic physicochemical quality (such as moisture, texture, color, and volatile substances) of aquatic products [9,10,11]. In addition, the effects of UHP on microorganisms and shelf life of aquatic products have also been evaluated [12,13]. Moreover, UHP treatment has significantly induced protein degradation, cross-linking, and oxidation of muscle when the pressure exceeds 100 MPa [1,14]. However, there is need for more studies on the effects of UHP on endogenous enzymes of aquatic products during storage [15]. Moreover, the correlation between endogenous enzyme activities, protein properties, and quality characteristics of *L. vannamei* remains highly fragmentary after treatment with UHP and later storage. This is because the quality characteristics of the aquatic products are greatly affected by pressure treatment methods and varieties.

Therefore, the objective of the current study was to explore the effects of UHP (100–500 MPa for 5 min) on texture and chromatic characteristics, protein properties (surface hydrophobicity, myofibril fragmentation index (MFI), trichloroacetic acid (TCA)-soluble peptides, carbonyl, dityrosine, free sulfhydryl, and disulfide bond) of iced shrimp. Further, the cathepsin (B, D, H, and L) activities, the mitochondrial membrane permeability, and the lysosomal membrane stability of the *L. vannamei* during 15 days storage at 0 °C were also evaluated. In addition, the correlation between the studied characteristics of iced shrimp treated with UHP were analyzed.

## 2. Materials and Methods

### 2.1. Sample Preparation and Ultra-High Pressure Treatment

Fresh shrimps (weight: 18.8 ± 2.3 g; length: 15.3 ± 0.8 cm) were purchased from the local market (Changchun, Jilin, China). Shrimps were placed in foam boxes containing crushed ice and transported to the laboratory within 12 h. The shrimps were rinsed with distilled water at 4 ± 0.5 °C and placed on drain pans with crushed ice for 5 min. After drying, shrimps were put into polyethylene bags and vacuum-packed.

The shrimps were randomly divided into six groups, five of which were processed at 100, 200, 300, 400, and 500 MPa for 5 min, respectively. An HPP L2-600/2 processor was used to conduct the UHP treatment under the following parameters: chamber volume, 2 L; diameter, 80 mm; depth, 400 mm; power, 5.5 kW; pump station pressure, 80 MPa; temperature, 25 °C. The parameters applied for the different pressure treatments were as shown in Appendix A. The UHP groups were later stored in a 0 °C refrigerator and covered with ice to minimize the temperature difference. Furthermore, the last group was directly stored as the control. All samples were stored for 15 days. The temperature was monitored and was seasonably supplied with ice during storage. Shrimps were analyzed after every 3 days and shelled before testing.

### 2.2. Texture Analysis

The surface of the second abdominal segment of shrimp (after decapitation) was used for textural profile analysis (TPA) testing by a texture analyzer (TA. XT. Plus, Stable micro systems, Godalming, UK) equipped with a P/5 cylindrical probe. The pre-test, test, and post-test speeds were 1 mm/s, 0.50 mm/s, and 1 mm/s, respectively. Meanwhile, the compression ratio was 50%, and the trigger force was 2 g.

The junction of the third and fourth abdominal segments of shrimp was used for shear force testing by the texture analyzer with a Warner–Bratzler blade. The pre-test and test speeds were 1 mm/s and 0.5 mm/s. In addition, the initial height was 15 cm, and the shear distance was 13 cm. The maximum force was recorded as the shear force value.

### 2.3. Chromatic Characteristics Measurement

The surface chromatic characteristics of the first abdominal segment of shrimp (after decapitation) were detected using a colorimeter (CM 5, Konica Minolta, Singapore). The parameters were set as follows: type, reflection measurement; aperture, Ø 8 mm; standard observer, 10°; illuminant, D65. The L* (luminance), a* (+a* means the redness while −a* the greenness), and b* (+b* means the yellowness while −b* the blueness) values were detected. Moreover, the *W* (whiteness) and Δ*E* (color difference) values were calculated according to the following equation.

The *W* value was calculated as the following equation:(1)$$W=100-{\left[{\left(100-L^{*}\right)}^{2}+{\left(a^{*}\right)}^{2}+{\left(b^{*}\right)}^{2}\right]}^{1/2}$$

The Δ*E* value was calculated as the following equation:(2)$$\Delta E={\left[{\left(L1^{*}-L0^{*}\right)}^{2}+{\left(a1^{*}-a0^{*}\right)}^{2}+{\left(b1^{*}-b0^{*}\right)}^{2}\right]}^{1/2}$$

Note: L0*, a0*, and b0* were original shrimp colorimetric values; L*, a*, and b* were the colorimetric values of shrimp samples after UHP.

### 2.4. Extraction of Myofibrillar Protein

The myofibrillar protein (MP) was extracted as the previously described method with some modifications [16]. A total of 2 g of muscle was homogenized (T18, I.K.A., St Neots, Germany) with 10 mL precool phosphate buffer solution (PBS) A (0.1 M NaCl, 2 mM MgCl_2_, 1 mM EGTA, 20 mM Na_2_HPO_4_/NaH_2_PO_4_, pH 7.0) at 12,000 rpm for 30 × 2 s. The homogenate was then centrifuged (Allegra X-30R, Beckman Coulter, Brea, CA, USA) at 5000× *g* for 10 min at 4 °C. The precipitate was mixed with five volumes of PBS A and centrifuged again. The operation was repeated for three times and the final pellets were dissolved in precool PBS B (0.6 M NaCl, 20 mM Na_2_HPO_4_/NaH_2_PO_4_, pH 7.0). The biuret method was used to measure concentration of the MP solution [17], and the final protein concentration was adjusted to 5 mg/mL.

### 2.5. Surface Hydrophobicity

Surface hydrophobicity was measured according to the previous report with minor modifications [16]. About 1 mL of MP (distilled water as blank) was mixed with 40 μL of bromophenol blue (BPB) solution (1 mg/mL). The mixture solution was vibrated at room temperature for 10 min and centrifuged at 10,000× *g* for 5 min at 4 °C. The absorbance of the supernatant was measured at 595 nm using a microplate analyzer (FLUOstar Omega, BMG LABTECH, Ortenberg, Germany). The result was calculated as the following equation:(3)$$\mathrm{BPB}\;\mathrm{bound}\left(\mu g\right)=\frac{40\;\mu g\;\times \left(\mathrm{OD}\;\mathrm{blank}\;-\;\mathrm{OD}\;\mathrm{sample}\right)}{\mathrm{OD}\;\mathrm{blank}}$$

### 2.6. Myofibril Fragmentation Index

The myofibril fragmentation index (MFI) was determined according to the previous method [18]. Briefly, the MP concentration was adjusted to 0.5 mg/mL with PBS B, and the absorbance was measured at 540 nm. The MFI was obtained by the OD value multiplied by 200.

### 2.7. Trichloroacetic Acid-Soluble Peptides

The trichloroacetic acid (TCA)-soluble peptides content of MP were measured as the previous method [19]. A total of 2 g of muscle was homogenized with 18 mL 5% TCA (*m*/*v*) at 10,000 rpm for 30 s. After being stored at 4 °C for 30 min, the mixture was centrifuged at 5000× *g* for 10 min. The supernatant was used to measure the content of TCA-soluble peptides using the BCA Protein Quantitation Kit (Beyotime Biotechnology, Haimen, China), and the content was expressed as μmol tyrosine/g muscle.

### 2.8. Free Sulfhydryl and Disulfide Bond Content

The free sulfhydryl and disulfide bond contents of MP were evaluated as previously described methods with minor adjustments [1]. Briefly, 0.1 mL of the MP solution (5 mg/mL) was mixed with 0.9 mL of PBS_F_ (3 mM EDTA, 1% SDS, 0.2 M Tris-HCl, pH 8.0) and 0.1 mL of DTNB (10 mM). Then, the mixture was incubated at 4 °C for 1 h to detect the content of free sulfhydryl. In addition, 0.1 mL of MP solution (5 mg/mL) was mixed with 1 mL of PBS_d_ (8 M urea, 3 mM EDTA, 1% SDS, 0.1 M Na_2_SO_3_, 1% NTSB, 0.2 M Tris-HCl, pH 9.5) and incubated in the dark at 25 °C for 25 min to detect the content of disulfide bond. The absorbance of mixture was measured at 412 nm and the contents were expressed as nmol/g protein.

### 2.9. Carbonyl Content

The carbonyl content was determined as previous description with minor modifications [20]. About 0.5 mL of MP solution (10 mg/mL) was mixed with 2 mL of 10 mM 2,4-dinitrophenylhydrazine (DNPH, dissolved in 2 N HCl; DNPH was replaced by 2 N HCl as blank) and incubated at 25 °C for 1 h. Then, 1.5 mL of 20% TCA (*w*/*v*) was added to the mixture and centrifuged at 11,000× *g* for 3 min. The precipitate was then washed three times with 2 mL of ethyl acetate/absolute ethanol. Further, the protein pellets were collected and resuspended in 6 M guanidine hydrochloride (dissolved in 20 mM PBS, pH 2.3), and centrifuged again. The absorbance of supernatant was measured at 370 nm using a Lambda 365 UV-VIS spectrophotometer (PerkinElmer Enterprise Management Co., Ltd., Shanghai, China). The content was calculated as the following equation:(4)$$\mathrm{Carbonyl}\;\mathrm{content}\left(\frac{\mathrm{nmol}}{\mathrm{mg}}\mathrm{protein}\right)=\frac{\Delta A370}{b\times 22000\times C}\times 10^{6}$$
where ΔA370—under 370 nm absorbance minus the blank absorbance; b—the light transmission distance at the 200 μL dosage per well, that is 0.588 cm; 22000—the molar extinction coefficient (m^−1^ cm^−1^); C—the protein concentration (mg/mL).

### 2.10. Dityrosine Content

The determination of dityrosine content was slightly adjusted based on the previous method [21]. Briefly, the MP solution was diluted with PBS B to 1 mg/mL. The dityrosine content was measured using a fluorescence spectrophotometer (Lumina Fluorescence Spectrometer Thermo, Waltham, MA, USA) at the excitation and emission wavelengths of 325 nm and 420 nm, respectively. The bandwidth was set as 10 nm. The dityrosine content of MP was then expressed as the fluorescence intensity divided by protein concentration (arbitrary units, AU).

### 2.11. Cathepsin B, D, H, and L Activities

Cathepsin B, H, and L activities were determined as the previous method with minor adjustments [22]. The muscle was homogenized with three volumes of PBS (pH 7.6) at 12,000 rpm for 20 s and centrifuged at 10,000× *g* for 30 min at 4 °C. The supernatant was the crude enzyme solution. The corresponding substrates, reaction buffers, and stopping solution of cathepsins were shown in Appendix A. The activities of cathepsin B, H, and L were expressed as the increased fluorescence measured at the excitation and emission wavelengths of 360 nm and 460 nm, respectively. One enzyme activity unit (U) was defined as the amount of enzyme required to release 1 nmol of 7-amino-4-methylcoumarin (AMC) per min at 37 °C (1 nmol/min). The standard curve of AMC was shown in Appendix A.

The activity of cathepsin D was detected using a mixed buffer (2.5% bovine hemoglobin: 0.2 M citrate buffer = 1:1) as the previous method [23]. The crude enzyme solution was incubated with buffer for 1 h at 37 °C and 10% TCA (*W*/*V*) was added to terminate the reaction. After centrifugation (16,000× *g* for 5 min), the supernatant was measured at the UV absorbance of 280 nm. One enzyme activity unit (U) was defined as the absorbance value increased per gram of crude enzyme solution per min at 37 °C.

### 2.12. Mitochondrial Membrane Permeability

The permeability of mitochondrial membrane was measured as previous methods with some improvements [24,25]. About 2 g of muscle was homogenized with 20 mL of precool extractant A (250 mM Sucrose, 10 mM Tris-HCl, 1 mM EDTA, pH 7.4) at 12,000 rpm for 15 s, and centrifuged at 1500× *g* for 15 min at 4 °C. The supernatant was then centrifuged again at 12,000× *g* for 20 min and the precipitate was mitochondria. Mitochondrial solution (3 mg/mL) was diluted with extractant B, later mixed with nine volumes of extractant B (230 mM Mannitol, 70 mM Sucrose, 3 mM Hepes, pH 7.4), and incubated for 3 min at 25 °C. The absorbance was measured at 540 nm.

### 2.13. Lysosomal Membrane Stability

The lysosomes in muscle tissue were isolated by differential centrifugation. A total of 1 g of muscle was homogenized with 9 mL extractant A at 12,000 rpm for 15 s and centrifuged at 1000× *g* for 15 min at 4 °C. The supernatant was centrifuged again at 10,000× *g* for 20 min and was used to determine the activity of free cathepsin D (AF) and the precipitate was suspended in 0.1% Triton X-100 as well as incubated for 30 min at 37 °C. The supernatant was used after the third centrifugation to determine the activity of the binding cathepsin D (AB). The stability was calculated as:(5)$$\mathrm{Stability}=\frac{\mathrm{AF}}{\mathrm{AB}}\times 100\%$$

### 2.14. Total Volatile Basic Nitrogen (TVB-N) and pH

The TVB-N was detected with reference to a previous method [1]. The TVB-N content was expressed as mg/100g sample. Moreover, the mixture pH was measured using an ST3100 pH meter (OHAUS International Trading Co., Ltd., Shanghai, China).

### 2.15. Statistical Analysis

All experiments were repeated three times independently, and the test results were expressed as mean ± standard deviation (mean ± SD). Graphpad prism 8 software was used for statistical analysis and graphing. Two-way ANOVA was used to compare the differences between groups, *p* < 0.05 was considered statistically significant.

## 3. Results and Discussion

### 3.1. Changes in Quality Characteristics

Results of the current study showed that the L* and b* values of muscles were significantly increased whereas the a* value of muscles was significantly decreased in shrimp under pressure above 300 MPa compared with that of the control (Table 1; *p* < 0.01). Previous studies have shown that the increase of muscle luminance may be because of the loss of active pigment. Moreover, pressure-induced protein degeneration changed the light reflectivity on the muscle surface, which could be another reason for affecting the luminance [26,27]. Lin et al. [12] reported that the L* value of Asian hard clam was significantly increased by treatment with UHP (≥200 MPa) for 3 min. Moreover, the changes in values of a*and b* may be attributed to the degradation of carotenoid pigments such as astaxanthin, which is caused by the treatment with pressure above 300 MPa for 5 min [28]. Meanwhile, it has been found that myoglobin is oxidized to ferrous myoglobin and this could be another reason for the yellowing of shrimp [27]. It was found that the *W* and *ΔE* of muscles were significantly improved with the increase in pressure (especially above 300 MPa) (Table 1; *p* < 0.01), which could be closely associated with the change of luminance. Moreover, it was noted that there was no significant change in trend of color-related values in each group during the whole storage period (Table 1).

As shown in Appendix A, the texture properties of shrimp were notably affected by the pressure above 300 MPa, especially at 500 MPa. The hardness and shear force of muscle were also enhanced by pressure above 300 MPa (*p* < 0.001) compared with the control. Moreover, it was notable that the gumminess, chewiness, and resilience were significantly promoted by 500 MPa treatment (*p* < 0.001), which could be related to the protein denaturation and aggregation of the muscle tissue. Similar results have been shown by the study conducted by Kaur et al. [10], whose results indicated that the hardness, gumminess, and chewiness of black tiger shrimp was promoted by UHP (300–600 MPa for 3–15 min). Furthermore, the peak values of hardness, gumminess, chewiness, and resilience in the tested groups were observed on the 3rd day of storage. However, the peak values of shear force, springiness, adhesiveness, and cohesiveness were not identified. This may be associated to the inability of contractile muscles to relax because of the decomposition of adenosine triphosphate (ATP) after death of shrimp (Rigor mortis) [29,30]. The texture parameters were significantly improved by treatment with UHP compared with fresh shrimp, which ensured that the speed of texture softening was effectively delayed during storage. This was an indication that UHP-treated shrimp was more conducive to storage compared with fresh shrimp.

Overall, with the increase of pressure, the L*, *W*, and Δ*E* values of muscle increased while a* decreased, and the appearance of shrimp observed was brighter and whiter (Table 1). Moreover, the hardness, shear force, gumminess, chewiness, and resilience of muscle were also significantly increased at pressure above 300 MPa (Appendix A). It is worth noting that the quality characteristics of the shrimp treated within 200 MPa were almost similar to those of fresh shrimp. Kung et al. [31] demonstrated that the lightness and texture values of milkfish flesh were increased significantly with the pressure increasing from 200 to 600 MPa. The author evaluated that 200 MPa for 5 min on milkfish flesh is a critical pressure for acceptably visual and physical properties compared to other higher-pressure groups. Since the sales market of the product is directly affected by its sensory characteristics, 300 MPa for 5 min is an acceptable critical pressure condition for shrimp processing.

### 3.2. Changes in Protein Properties

#### 3.2.1. Protein Degradation

Surface hydrophobicity is one of the leading indicators for detection of the changes in protein structure. Usually, the hydrophobic groups of natural protein are buried inside. The BPB is used to monitor the changes in conformation of MP by binding to its hydrophobic sites [32]. As shown in Figure 1A, it was evident that the surface hydrophobicity of fresh shrimp was 5.14 μg. UHP (especially above 400 MPa) significantly increases the hydrophobic value of muscle due to it exposing the protein to more hydrophobic sites by changing the native conformation of MP. Similarly, previous studies have shown that the surface hydrophobicity index (SHI) of cod protein was significantly increased at 200 MPa/min at 25 °C for 20 min [14]. During storage, it was found that the value of the changes in protein structure was increased by 2.61-, 2.06-, 1.83-, 1.29-, 0.57-, and 0.44-folds, respectively. These results showed that UHP treatment significantly slowed down the protein degradation process.

The MFI value reflects the structural integrity of MF [33]. It was found that the MFI was significantly enhanced by pressure above 400 MPa compared with the control (*p* < 0.001; Figure 1B), suggesting that UHP treatment damaged the MP integrity. Moreover, the MFI in each tested group was increased with the storage period (*p* < 0.01) and this suggested that there was further damage in the structure of MF.

The TCA-soluble peptides include endogenous peptides and the degradation products that accumulate during storage [34]. According to Figure 1C, it was evident that the TCA-soluble peptides content in all treated groups of shrimp had no significant difference in the first 3 days of storage. However, the content was notably prevented by the pressure from the 6th day (*p* < 0.05). In addition, the content in each group was increased from 0.268, 0.273, 0.282, 0.292, 0.277, and 0.325 μmol Tyr/g on day 0 to 1.134, 1.032, 0.937, 0.844, 0.708, and 0.585 μmol Tyr/g on day 15, respectively. This indicated that the accumulation of the protein degradation products in muscle was effectively inhibited by treatment with UHP during storage. The observed trends were in agreement with the results of a study conducted by Chen et al. [1], who showed that the content of MFI and TCA-soluble peptides in *Penaeus monodon* was significantly increased by treatment with 500 MPa for 5 min and during storage, which implied that the structural integrity of the protein was directly disrupted.

#### 3.2.2. Protein Oxidation

One of the phenomena of protein oxidation is the conversion of its sulfhydryl groups to disulfide bonds. It is worth noting that sulfhydryl groups can significantly affect the functional properties of proteins [35]. As showed in Figure 2A, it was found that the content of free sulfhydryl of MP was significantly increased by treatment with UHP, which may be due to protein denaturation, resulting in the free sulfhydryl being exposed from the inside of the molecules. A previous study has shown that there was a significant increase in the surface sulfhydryl groups of tilapia actomyosin after being treated with pressure between 100 and 300 MPa for 10 min [36]. However, it was evident that the content of disulfide bond was not significantly changed by pressure even at 500 MPa for 5 min compared to the control (Figure 2B), and the results were consistent with that of previous study [1]. The disulfide bond cannot be directly broken or formed by the pressure of 500 MPa [37]. During storage, the content of free sulfhydryl in the tested groups reached the maximum on the 6th day (70.78, 81.89, 83.96, 94.80, 109.08, and 119.06 nmol/g, respectively) and then was down-regulated. This may be attributed to the free sulfhydryl groups being bound in varying degrees, resulting in the oxidation of protein in the later period.

The content of carbonyl is considered a general index to evaluate the degree of protein oxidation in meat. Carbonyl is formed by breaking the amino acid side chains and peptide chains vulnerable to free radical attack [20]. Results of the present study showed that the content of carbonyl during storage in each group was increased from 1.53, 1.61, 1.59, 1.68, 2.00, and 2.66 nmol/g protein to 247.00, 239.04, 227.27, 209.46, 163.47, and 121.61 nmol/g of protein, respectively (Figure 2C). It was evident that pressure above 300 MPa effectively slowed down accumulation of the carbonyl, which could be because of delayed destruction of amino acid side chains as well as the peptide chains caused by UHP during storage.

Dityrosine is formed through the mutual complexation of two oxidized tyrosine residues. The content of dityrosine showed the degree of crosslinking for the oxidized protein [21]. According to Figure 2D, it was evident that the content of dityrosine in each group was notably enhanced through prolongation of the storage period (*p* < 0.001) and hence illustrating that protein oxidation occurred in all groups. Remarkably, there was a negative correlation between the pressure and accumulation of dityrosine after the 6th day of storage (*p* < 0.01). This suggested that the protein oxidation in muscle was effectively retarded by treatment with UHP and hence the shrimp quality was maintained during storage.

In this study, treatment with UHP above 300 MPa significantly increased the values of surface hydrophobicity, MFI, free sulfhydryl, and dityrosine (Figure 1 and Figure 2). This suggested that UHP treatment disrupted the structural integrity of protein, promoted the exposure of internal hydrophobic sites and conformational changes in the protein. During storage, UHP treatment delayed the accumulation of TCA-soluble peptides, carbonyl, and dityrosine (Figure 1 and Figure 2). This indicated that the protein degradation and oxidation of shrimp was effectively delayed during storage. Zeng et al. [38] found that the protein degradation and oxidation were significantly retarded after the Yesso scallop was treated at higher pressure (400 and 500 MPa for 5 min). Moreover, because of the specificity of protein function, oxidative modifications can directly cause a variety of changes in food quality properties, such as hydrophobicity, solubility, color, texture, and digestibility [39,40]. The significant delay of the protein oxidation of shrimp was caused by treatment with UHP during storage, which may alleviate the quality characteristics deterioration of stored shrimp.

### 3.3. Changes in Endogenous Enzyme Activities

Remarkably, it was found that the activities of cathepsin B, D, H, and L were first increased and then decreased with an increase in pressure (Figure 3A–D). This could be because of the spatial structure of binding groups in the enzyme active center were changed by exposure to pressure of 300 MPa. The change in the spatial structure of binding groups promoted the combination of enzyme and substrate, and hence accelerated the enzymatic reaction [41]. However, it was evident that the spatial structure of enzymes was irreversibly destroyed by higher pressure, resulting in the loss of enzyme activities (*p* < 0.001) [42]. Moreover, it was noted that cathepsin B was more sensitive to pressure than other endogenous enzymes, and its activity was decreased by 27.26% and 43.42% after treatment with pressure at 400 and 500 MPa, respectively (Figure 3A). Furthermore, the initial activity of cathepsin D was higher than that of the other enzymes whereby the activity was down-regulated by 22.56% after treatment with 500 MPa (Figure 3B). The results were in agreement with the findings of Yu et al. [43], who found that the activities of cathepsins B and D were only inhibited under higher pressures (≥400 MPa) for 15 min in crude enzyme extracts prepared from grass carp fillets. Activities of the four cathepsin in all groups showed an inverted “V” change trend in varying degrees during storage. It was evident that the higher the pressure, the more the activities were significantly inhibited (*p* < 0.01) and the trend was in agreement with finding of Yu et al. [43]. Generally, it was evident that the activities of cathepsin were effectively inhibited by UHP (above 300 MPa). Therefore, it was noted that protein degradation and texture deterioration that are caused by endogenous enzymes in shrimp were retarded during storage period.

It has been found that mitochondria play an essential role in apoptosis and necrosis [44]. Only small molecular substances are normally allowed to freely pass through the mitochondrial membrane. However, the permeability of mitochondrial membrane is significantly improved by apoptosis hence increasing material exchange [45]. As shown in Figure 4A, it was found that an increase in pressure reduced the absorbance of mitochondrial suspension from 0.34 to 0.26 (*p* < 0.001). It was evident that protein denaturation and apoptosis were triggered by UHP, which significantly increased the permeability of mitochondrial membrane. Furthermore, the absorbance suspension in all groups showed a downward trend during storage, indicating that the integrity of mitochondrial membrane was continuously destroyed.

Results of the present study showed that with the increase of pressure, the stability of lysosomal membrane was first increased by 0.1% (at 200 MPa) and then decreased (*p* < 0.001) (Figure 4B). Moreover, the lysosomal membrane stability in each group was reduced by 33.21, 38.44, 45.85, 44.80, 45.09, and 45.30%, respectively (*p* < 0.001), on the 3rd day of storage. Then, lysosomal membrane stability in each group showed a flat downward trend (Figure 4B).

Proteolysis and texture softening are the typical changes that occur during storage of shrimp products. They are closely related to various endogenous enzymes such as cathepsin and calpain in muscle tissue [46]. Various cathepsins have been found in lysosomes to participate in cell metabolism. In dead shrimp, the stability of lysosomal membrane may gradually decrease, and the cathepsins are then released into cells [47]. In this study, treatment with UHP caused change in the permeability of mitochondrial membrane, which induced apoptosis, destroyed the stability of lysosomal membrane, and promoted release of the cathepsins in lysosome of shrimp [48]. However, it was evident that pressure caused protein denaturation and hence the cathepsin (B, D, H, and L) activities were inhibited. Finally, the protein denaturation and texture deterioration of shrimp induced by endogenous enzymes were significantly retarded by treatment with UHP above 300 MPa during storage.

### 3.4. Changes in TVB-N and pH

As showed in Figure 5A, the pH of the shrimp treated with 100, 200, and 300 MPa (6.69, 6.72, and 6.75, respectively) was closer to that of the shrimp in the control group (6.67). However, the pH was enhanced with treatment above 400 MPa (6.92 and 7.14, respectively, *p* < 0.001). The results also showed that the exposure of alkaline amino acid residues was promoted by UHP [11]. The findings of the current study were in consonance with the results of Ramirez et al. [49] who studied the effect of UHP on the pH of grass carp fillets and albacore tuna. During storage, it was noted that the pH in all treated groups was enhanced by 3.90, 3.64, 3.77, 3.56, 1.73, and 1.45%, respectively. This could be because the protein was continuously decomposed by microorganisms such as psychrotrophic bacteria and coliform [12]. Furthermore, it was found that a large number of nitrogen-containing substances were simultaneously formed in the present study. However, the level of accumulation of the products from microbial decomposition process was significantly deferred by treatment with UHP (above 300 MPa).

The TVB-N refers to the sum of nitrogen-containing volatile substances (ammonia, dimethylamine, and trimethylamine) in the stored aquatic products. The TVB-N stems from the degradation of endogenous peptides and products of protein decomposition produced by the action of endogenous enzymes and bacteria [34,50]. According to Figure 5B, the TVB-N in all groups of shrimp showed an upward trend during the whole storage period. Compared with the initial level, the TVB-N of shrimp was promoted by 16.75, 13.14, 8.80, 5.89, 4.06, and 3.07-fold, respectively. UHP treatment effectively inhibited accumulation of TVB-N. Moreover, it was found that the TVB-N in the control group reached 28.18 mg/100 g at day 6, which was almost close to the national standard limit for fresh aquatic products (30 mg/100 g). Furthermore, the TVB-N of the shrimp treated with 100 and 200 MPa reached 32.35 and 28.94 mg/100 g at the 9th day, respectively. After being treated with pressure above 300 MPa, the TVB-N in shrimp did not exceed the limit standard at day 15 (28.73, 21.56, and 16 58 mg/100 g, respectively). The data showed that the activities of cathepsin (B, D, H, and L) and accumulation of the protein decomposition products during storage were significantly inhibited by UHP. It was worth noting that the shelf life of *L. vannamei* was prolonged by at least 8 days after being treated with pressure above 300 MPa for 5 min (Figure 5B). Kaur et al. [51] also found that treatment with pressure at 435 MPa for 5 min had the highest impact in reducing TVB-N produced in muscle tissue of black tiger shrimp compared with pressure at 100 and 270 MPa for 5 min. In addition, the study showed that the shelf life of shrimp treated at 435 MPa for 5 min was prolonged by 10 days during storage at 2 ± 0.5 °C compared with that of the untreated samples.

## 4. Conclusions

UHP treatment within 300 MPa for 5 min can maintain the natural qualities of shrimp, such as color and texture, and thus relatively prolong their freshness. UHP treatment above 300 MPa changed the natural protein structure of shrimp and caused protein degradation and oxidation, resulting in significant differences in its quality characteristics compared with fresh shrimp. During storage, UHP treatment increased the permeability of mitochondrial membrane whereas decreased the stability of lysosomal membrane of shrimp. Subsequently, cathepsins B, D, H, and L were released but the enzyme activities were significantly inhibited by pressure above 300 MPa. Hence, the deterioration of protein properties and texture characteristics induced by cathepsins in shrimp deferred during storage. Generally, UHP treatment with 300 MPa for 5 min effectively maintained the stability of protein properties and quality characteristics, and prolonged the shelf life of stored *L. vannamei* by 8 days. The results of this study might be helpful in understanding the effects of UHP treatment on the endogenous enzymes, protein properties and quality characteristics of *L. vannamei* during iced storage.

## Figures and Tables

**Figure 1 molecules-27-06302-f001:**
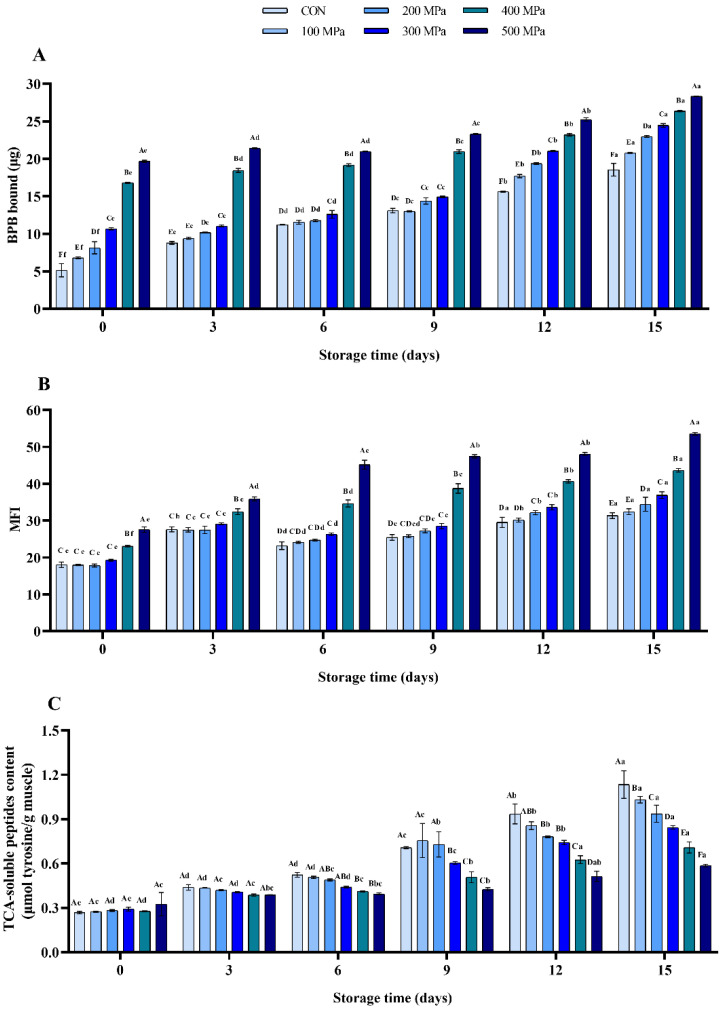
Effects of UHP treatment on protein degradation of shrimp during storage: the content of (**A**) surface hydrophobicity, (**B**) MFI, and (**C**) TCA-soluble peptides.

**Figure 2 molecules-27-06302-f002:**
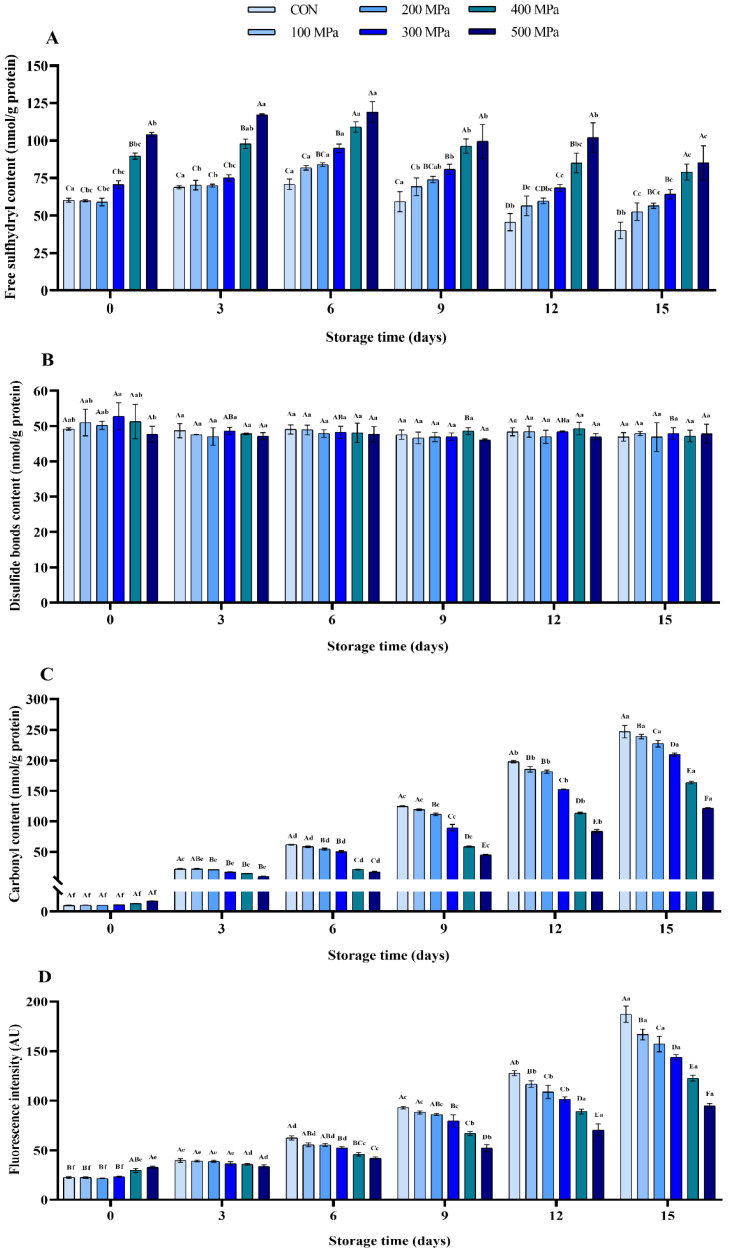
Effects of UHP treatment on protein oxidation of shrimp during storage: the content of (**A**) free sulfhydryl, (**B**) disulfide bond, (**C**) carbonyl, and (**D**) dityrosine.

**Figure 3 molecules-27-06302-f003:**
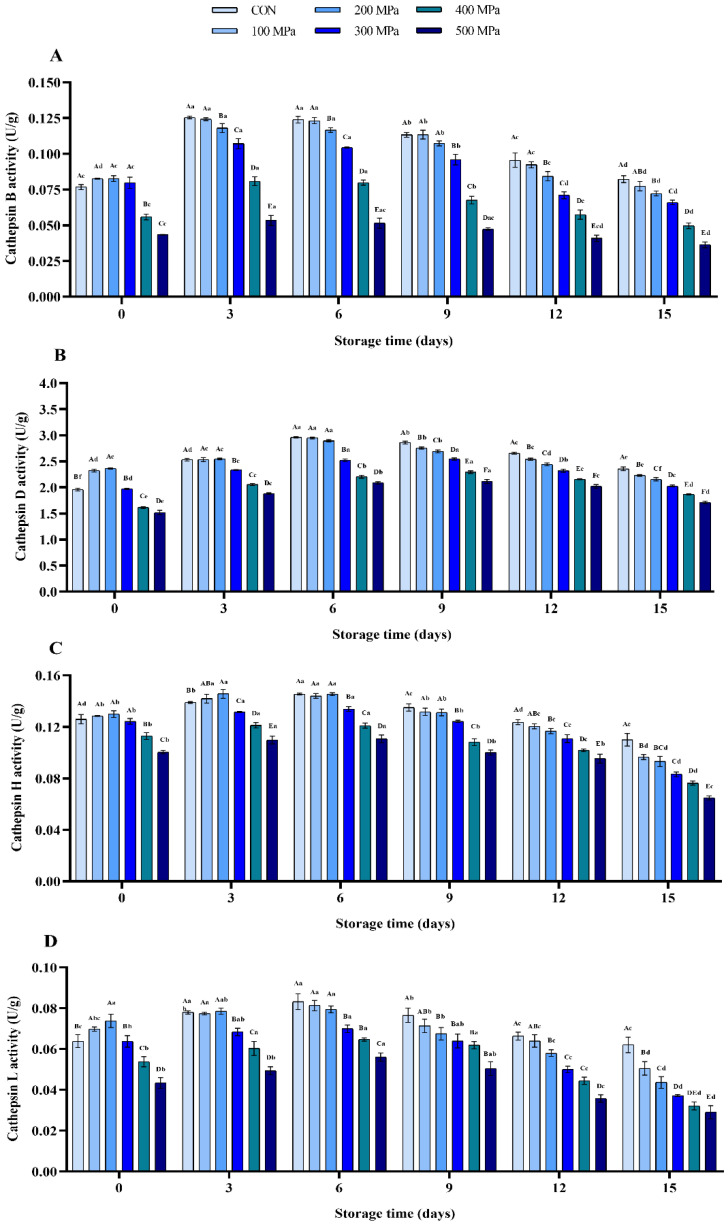
Effects of UHP treatment on the endogenous enzyme activities of shrimp during storage: (**A**) cathepsin B, (**B**) cathepsin D, (**C**) cathepsin H, and (**D**) cathepsin L.

**Figure 4 molecules-27-06302-f004:**
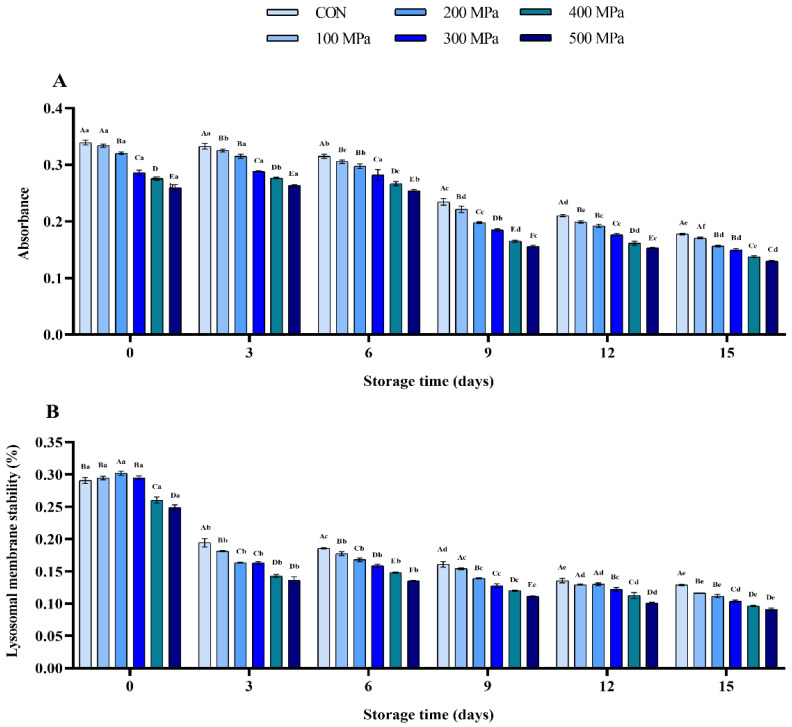
Effects of UHP treatment on the (**A**) mitochondrial suspension absorbance and (**B**) lysosomal membrane stability of shrimp during storage.

**Figure 5 molecules-27-06302-f005:**
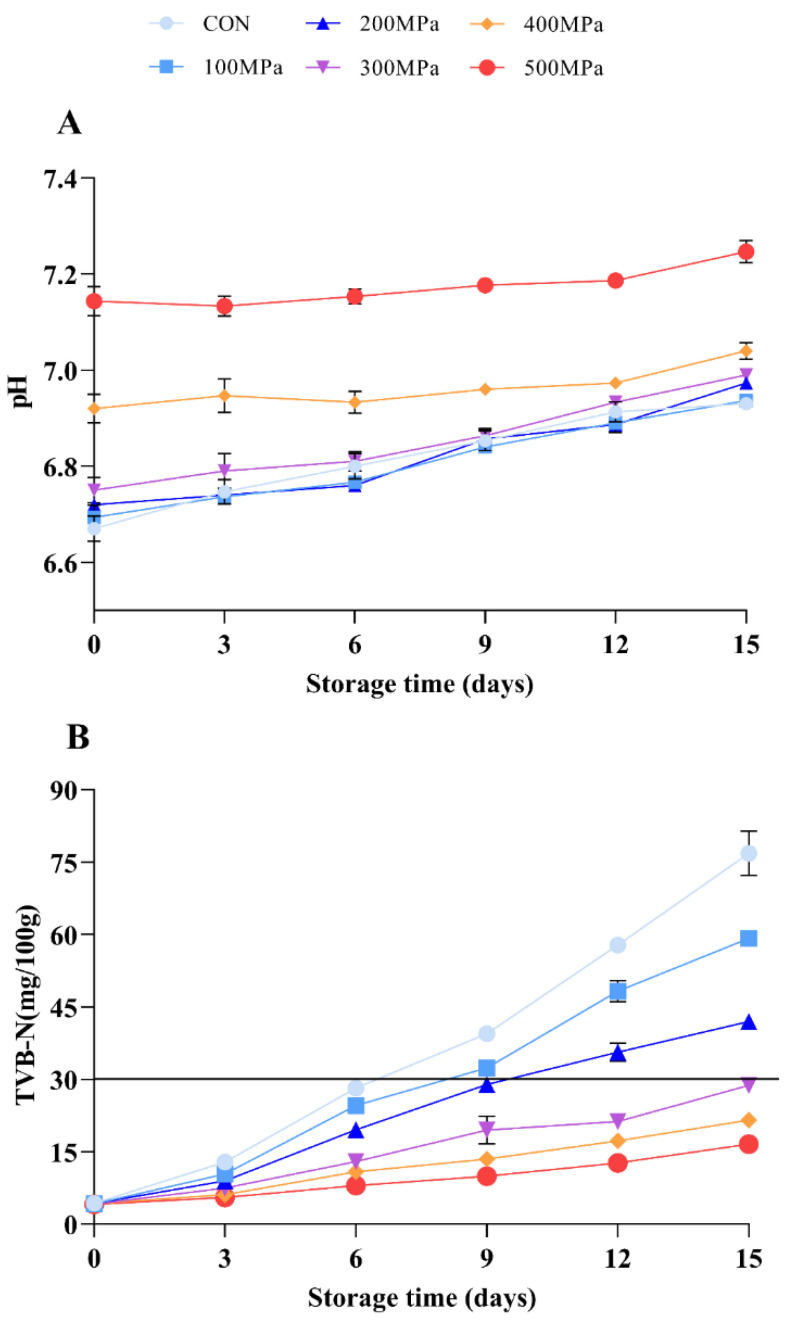
Effects of UHP treatment on the (**A**) pH and (**B**) TVB-N of shrimp during storage. Solid-line represents 30 mg/100 g of TVB-N, the national standard limit for fresh aquatic products.

**Table 1 molecules-27-06302-t001:** Effects of UHP treatment on color changes of shrimp during storage.

	Pressure Treatment	Storage Time (days)					
		0	3	6	9	12	15
L*	Control	40.7 ± 2.49 ^Ec^	45.48 ± 1.19 ^Fb^	46.95 ± 2.64 ^Dab^	50.44 ± 0.46 ^Ea^	50.42 ± 2.09 ^Ba^	47.22 ± 0.96 ^Eab^
	100 MPa	46.09 ± 1.14 ^Db^	49.38 ± 2.89 ^EFb^	49.22 ± 0.93 ^CDab^	53.03 ± 0.14 ^DEa^	51.97 ± 1.24 ^BCa^	52.26 ± 0.19 ^Da^
	200 MPa	50.08 ± 1.16 ^CDd^	50.43 ± 2.31 ^DEcd^	53.72 ± 1.79 ^BCabcd^	53.1 ± 0.22 ^CDEbcd^	56.16 ± 0.53 ^Bab^	58.28 ± 2.52 ^Ca^
	300 MPa	51.56 ± 1.58 ^Ce^	54.95 ± 3.38 ^CDcde^	54.24 ± 2.43 ^Bde^	64.58 ± 1.42 ^Ba^	56.44 ± 1.28 ^Bbcd^	63.81 ± 1.14 ^Ba^
	400 MPa	65.84 ± 2.56 ^Bde^	65.11 ± 3.95 ^Bcde^	68.24 ± 3.54 ^Acde^	75.42 ± 3.39 ^Aa^	69.04 ± 0.96 ^Abcde^	71.64 ± 1.6 ^Aabcd^
	500 MPa	79.64 ± 2.99 ^Aa^	74.32 ± 0.6 ^Abcd^	72.56 ± 1.18 ^Acd^	77.67 ± 0.95 ^Aab^	71.82 ± 1.67 ^Ad^	75.18 ± 1.15 ^Aabcd^
a*	Control	1.19 ± 0.29 ^BCab^	−0.03 ± 0.16 ^Cbcde^	−0.34 ± 0.88 ^Bde^	−0.87 ± 0.61 ^Cde^	1.89 ± 0.23 ^Aa^	−0.31 ± 0.23 ^BCcde^
	100 MPa	0.03 ± 0.2 ^Cab^	0.26 ± 0.24 ^BCa^	−0.47 ± 0.56 ^Babc^	−1.21 ± 0.5 ^CDbc^	0.74 ± 0.75 ^Aa^	−1.39 ± 0.99 ^Cc^
	200 MPa	0.27 ± 0.29 ^Cabcd^	0.15 ± 0.38 ^BCbcd^	0.63 ± 0.96 ^Babc^	0.08 ± 0.31 ^ACcd^	1.54 ± 0.43 ^Aa^	−0.96 ± 0.47 ^Cd^
	300 MPa	2.26 ± 0.51 ^Aba^	2.24 ± 0.53 ^Aa^	0.69 ± 0.66 ^Bbcd^	−0.2 ± 0.14 ^ACd^	0.75 ± 0.35 ^Abcd^	0.6 ± 0.56 ^ABcd^
	400 MPa	2.66 ± 0.82 ^Aa^	2.84 ± 0.77 ^Aa^	2.48 ± 0.42 ^Aa^	−0.4 ± 0.21 ^Bb^	1.71 ± 0.24 ^Aa^	1.91 ± 0.56 ^Aa^
	500 MPa	−2.67 ± 0.17 ^Da^	−2.75 ± 0.44 ^Da^	−2.14 ± 0.78 ^Ca^	−2.41 ± 0.62 ^Da^	−2.14 ± 1.04 ^Ba^	−2.98 ± 0.23 ^Da^
b*	Control	−1.17 ± 0.8 ^CDEabc^	−0.91 ± 0.83 ^DEabc^	−0.39 ± 1.59 ^CDEabc^	−3.09 ± 0.12 ^Dc^	0.67 ± 1.76 ^BCDa^	−3 ± 1.03 ^Cbc^
	100 MPa	−2.37 ± 0.37 ^DEab^	0.05 ± 1.92 ^BCDEa^	−1.6 ± 2.08 ^Eab^	−2.74 ± 1.35 ^CDab^	−1.61 ± 1.42 ^Dab^	−3.93 ± 0.6 ^Cb^
	200 MPa	−2.56 ± 0.85 ^Ea^	−0.96 ± 0.95 ^Ea^	−0.12 ± 1.89 ^BCDEa^	−1.58 ± 0.82 ^CDa^	−0.76 ± 1.23 ^CDa^	−2.6 ± 0.87 ^BCa^
	300 MPa	2.43 ± 0.53 ^ABabc^	2.07 ± 0.58 ^ABabc^	1.91 ± 1.14 ^BCDabc^	1.35 ± 0.81 ^ABbc^	4.46 ± 1.5 ^Aa^	0.08 ± 0.08 ^ABc^
	400 MPa	4.76 ± 1.63 ^Aabc^	4.46 ± 0.95 ^Aabc^	7.1 ± 1.65 ^Aa^	4.05 ± 0.61 ^Abc^	4.73 ± 1.48 ^Aabc^	2.89 ± 1.06 ^Ac^
	500 MPa	0.44 ± 1.33 ^BCDa^	−0.01 ± 1.46 ^CDEa^	−0.63 ± 1 ^DEa^	−0.81 ± 1.54 ^BCDa^	−1.29 ± 1.02 ^CDa^	−0.12 ± 0.31 ^Ba^
*W*	Control	40.68 ± 2.5 ^Ec^	45.47 ± 1.19 ^Eb^	46.92 ± 2.63 ^Eab^	50.34 ± 0.47 ^Ca^	50.36 ± 2.1 ^Ca^	47.13 ± 0.95 ^Eab^
	100 MPa	46.04 ± 1.13 ^Db^	49.36 ± 2.87 ^DEab^	49.16 ± 0.87 ^DEab^	52.92 ± 0.11 ^Ca^	51.92 ± 1.29 ^BCa^	52.07 ± 0.19 ^Da^
	200 MPa	50.01 ± 1.19 ^CDc^	50.41 ± 2.32 ^CDc^	53.68 ± 1.81 ^CDabc^	53.07 ± 0.2 ^Cbc^	56.11 ± 0.54 ^Bab^	58.18 ± 2.57 ^Ca^
	300 MPa	51.44 ± 1.57 ^Cc^	54.84 ± 3.34 ^Cbc^	54.18 ± 2.4 ^Cbc^	64.55 ± 1.44 ^Ba^	56.19 ± 1.33 ^Bb^	63.8 ± 1.14 ^Ba^
	400 MPa	65.36 ± 2.28 ^Bc^	64.68 ± 3.75 ^Bc^	67.31 ± 3.14 ^Bbc^	75.09 ± 3.44 ^Aa^	68.61 ± 0.98 ^Abc^	71.41 ± 1.58 ^Aab^
	500 MPa	79.42 ± 2.89 ^Aa^	74.14 ± 0.64 ^Abc^	72.45 ± 1.24 ^Ac^	77.49 ± 1.02 ^Aab^	71.69 ± 1.76 ^Ac^	75 ± 1.13 ^Aabc^
Δ*E*	Control	−	−	−	−	−	−
	100 MPa	5.69 ± 1.77 ^Da^	4.09 ± 1.9 ^Da^	3.79 ± 1.27 ^CDa^	2.88 ± 0.36 ^Ca^	3.36 ± 2.34 ^BCa^	5.47 ± 0.56 ^Da^
	200 MPa	9.43 ± 1.39 ^CDab^	5.07 ± 1.03 ^Dc^	7.13 ± 0.55 ^BCbc^	3.35 ± 0.53 ^Cc^	6.09 ± 2.02 ^Bbc^	11.15 ± 1.67 ^Ca^
	300 MPa	11.52 ± 0.83 ^Cbc^	10.25 ± 2.17 ^Ccd^	7.79 ± 0.68 ^Bcd^	14.87 ± 0.68 ^Bab^	7.23 ± 0.75 ^Bd^	16.91 ± 1.16 ^Ba^
	400 MPa	25.93 ± 1.96 ^Ba^	20.57 ± 3.43 ^Bb^	22.81 ± 1.33 ^Aab^	26.02 ± 3.6 ^Aa^	19.25 ± 1.24 ^Ab^	25.27 ± 1.18 ^Aa^
	500 MPa	39.2 ± 3.19 ^Aa^	28.99 ± 1.16 ^Ab^	25.78 ± 1.76 ^Abc^	27.41 ± 0.86 ^Ab^	21.97 ± 1.2 ^Ac^	28.24 ± 0.21 ^Ab^

Different capital letters in the same column indicate significantly different (*p* < 0.05). Different small letters in the same row indicate significantly different (*p* < 0.05).

## Data Availability

Not applicable.

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
