# Peer review of "Effects of Ultra-High Pressure on Endogenous Enzyme Activities, Protein Properties, and Quality Characteristics of Shrimp (Litopenaeus vannamei) during Iced Storage"

_molecules, 2022, doi:10.3390/molecules27196302_

Round 1

Reviewer 1 Report (Previous Reviewer 1)

The authors have addressed my concerns in the present revision. The modifications and updates performed significantly improved the quality of the manuscript.

Author Response

Comment : The authors have addressed my concerns in the present revision. The modifications and updates performed significantly improved the quality of the manuscript.

Response: We appreciate the reviewer very much for the valuable comments and suggestions on our manuscript entitled “Effects of Ultra-high Pressure on Endogenous Enzyme Activities, Protein Properties, and Quality Characteristics of Shrimp (Litopenaeus Vannamei) During Iced Storage” (molecules-1911672). They effectively improved the quality of our manuscript.

Reviewer 2 Report (Previous Reviewer 2)

Effects of Ultra-high Pressure on Endogenous Enzyme Activities, Protein Properties, and Quality Characteristics of Shrimp (Litopenaeus Vannamei) During Iced Storage

 Manuscript is well organized: well written Introduction part, detailed Material and methods chapter, Results and discussion part contains comparison with results found in literature. Conclusion summarizes all results statistically evidenced. There is the reason I have only minor comments given in the following list.

- Line 54 correct unit is MPa not MP.

- There are given on several places of the manuscript text references of literature by first author family name but prescribed brackets are given at the end of the sentence. I prefer that brackets have to be placed directly after family name; see lines 221-222, 237-239, 280-283, 351-353, 397-398, 421-425 (first line number contains reference by family name, second line number contains brackets with correct reference number).

- Line 232 exchange “sheer” by “shear”.

 - Supplementary Table S3 contains parameters Hardness and Shear force in unit kilogram. SI system reserved kilogram for mass. Force has unit Newton.

Author Response

We would like to show our heartfelt thanks to the reviewers for their kind and helpful comments. According with these advice, we amended the relevant part in manuscript, and someone of the questions were answered below.

Comment 1: Line 54 correct unit is MPa not MP.

Response: Thanks for the reviewer's careful work. We have corrected the unit “MP” to “MPa”. (Page 2, Line 53)

Comment 2: There are given on several places of the manuscript text references of literature by first author family name but prescribed brackets are given at the end of the sentence. I prefer that brackets have to be placed directly after family name; see lines 221-222, 237-239, 280-283, 351-353, 397-398, 421-425 (first line number contains reference by family name, second line number contains brackets with correct reference number).

Response: Thanks for the reviewer's careful work. We have placed brackets after first author family name (Page 6, Line 219 and Line 235) (Page 8, Line 289) (Page 12, Line 362) (Page 14, Line 411) (Page 15, Line 435).

Comment 3: Line 232 exchange “sheer” by “shear”.

Response: Thanks for the reviewer's careful work. We have corrected the expression: “sheer” to “shear (Page 6, Line 230)

Comment 4: Supplementary Table S3 contains parameters Hardness and Shear force in unit kilogram. SI system reserved kilogram for mass. Force has unit Newton.

Response: Thanks for the reviewer's careful work. We have correct the units of Hardness and Shear force to “N” in Supplementary Table S3.

Reviewer 3 Report (Previous Reviewer 3)

Introduction

The authors have improved the introduction with the background and recent scientific developments in the subject under consideration, as published in the literature. It also now clearly outlines the issues that the researchers dealt with in their research. Therefore, I consider comment 1 to be sufficiently improved.

Materials and methods

The authors replied that they had performed the UHP experiment three times but nowhere did they write this in the article. Such information must be found in Section 2.1.

The authors have significantly improved the descriptions of the research methodologies used. They have improved them in terms of a more detailed description and in terms of language. The methodologies are now clearer and better presented. Therefore, I consider comment 2 to be sufficiently improved.

Results and discussion

The authors described their research results very well as it was in the previous version of the article. However, they still have not discussed their results with the results available in the literature on the effects of HHP/UHP on seafood quality. Therefore, I consider comment 3 is not sufficiently carried out.

Conclusions

The authors, in response to comment 4 on the conclusions section, state that they have added information in the results and discussion section. This is an inappropriate approach. The reader, after reading only the conclusions, should know what are the most important achievements of the published research without familiarizing himself with the entire article. Therefore, the conclusions section needs to be supplemented with at least two more research conclusions.

Author Response

We apologize for the extra work caused by the duplicate submission of the manuscript due to insufficient revision. Thank you again for providing detailed revisions to our manuscript, it was very helpful to us. We have further carefully revised the manuscript based on the comments. We would appreciate it if you would take the time to review our manuscript again. The reply to the revised comments is as follows:

Comment 1: Results and Discussion

The authors described their research results very well as it was in the previous version of the article. However, they still have not discussed their results with the results available in the literature on the effects of HHP/UHP on seafood quality. Therefore, I consider comment 3 is not sufficiently carried out.

(Comment 3: Results and Discussion

The results have been widely described, but there is no in-depth discussion, especially based on the literature on the effects of high pressure on seafood.)

Response: Thanks for the reviewer's careful work. We readjusted and supplemented the results discussion of the parameters in combination with relevant literature in section 3. Results and discussion.

  1. In section 3.1 Changes in quality characteristics

We added the description of the effect of UHP treatment on quality characteristics (color and texture values) of shrimp (Page 6, Line 246-251). We cited the study of Kung et al., who evaluated the appropriate pressure on milkfish flesh for acceptably visual and physical properties (Page 6, Line 251-255), and we discussed acceptable critical pressure condition (300 MPa for 5 min) for processing shrimp in our manuscript (Page 6, Line 255-257). The descriptions in the manuscript are as follows:Overall, with the increase of pressure, the L*, W, and ΔE values of muscle increased while a* decreased, and the appearance of shrimp observed was brighter and whiter (Table 1). Moreover, the hardness, shear force, gumminess, chewiness and resilience of muscle were also significantly increased at pressure above 300 MPa (Table S3). It is worth noting that the quality characteristics of the shrimp treated within 200 MPa were almost similar to those of fresh shrimp. Kung et al. [31] demonstrated that the lightness and texture values of milkfish flesh were increased significantly with the pressure increasing from 200 to 600 MPa. The author evaluated that 200 MPa for 5 min on milkfish flesh is a critical pressure for acceptably visual and physical properties compared to other higher-pressure groups. Since the sales market of the product is directly affected by its sensory characteristics, 300 MPa for 5 min is an acceptable critical pressure condition for shrimp processing.

  1. In section 3.2.2 Protein oxidation:

We added the description of the effect of UHP treatment on protein degradation and oxidation of shrimp, and found that UHP treatment above 300 MPa caused protein denaturation and degradation but delayed the deterioration of protein properties in shrimp during storage (Page 10, Line 331-337). We cited the study of Zeng et al, who found that the deterioration of protein properties in Yesso scallop was retarded after higher pressure treatment (Page 10, Line 337-339). In addition, based on the specificity of protein function, we discussed the potential impact of protein oxidative modifications on the quality characteristics of shrimp during storage (Page 10, Line 339-344). The descriptions in the manuscript are as follows:“ In this study, treatment with UHP above 300 MPa significantly increased the values of surface hydrophobicity, MFI, free sulfhydryl, and dityrosine (Figure 1-2). This suggested that UHP treatment disrupted the structural integrity of protein, promoted the exposure of internal hydrophobic sites and conformational changes in the protein. During storage, UHP treatment delayed the accumulation of TCA-soluble peptides, carbonyl, and dityrosine (Figure 1-2). This indicated that the protein degradation and oxidation of shrimp was effectively delayed during storage. Zeng et al. [38] found that the protein degradation and oxidation were significantly retarded after the Yesso scallop was treated at higher pressure (400 and 500 MPa for 5 min). Besides, because of the specificity of protein function, oxidative modifications can directly cause a variety of changes in food quality properties, such as hydrophobicity, solubility, color, texture, and digestibility [39,40]. The significant delay of the protein oxidation of shrimp was caused by treatment with UHP during storage, which may alleviate the quality characteristics deterioration of stored shrimp.

  1. Section 3.3 Changes in endogenous enzyme activities

We supplemented the description of the relationship between lysosomes, endogenous enzymes, proteolysis and texture softening (Page 12, Line 387-391). We found that UHP treatment above 300 MPa altered the permeability of mitochondrial membrane and the stability of lysosomal membrane, inhibited the activities of cathepsin, and ultimately delayed cathepsin-induced protein and texture deterioration of stored shrimp (Page 12, Line 391-398). The descriptions in the manuscript are as follows:“ Proteolysis and texture softening are the typical changes that occurs during storage of shrimp products. They are closely related to various endogenous enzymes such as cathepsin and calpain in muscle tissue [46]. Various cathepsins have been found in lysosomes to participate in cell metabolism. In dead shrimp, the stability of lysosomal membrane may gradually decrease, and the cathepsins are then released into cells [47]. In this study, treatment with UHP caused change in the permeability of mitochondrial membrane, which induced apoptosis, destroyed the stability of lysosomal membrane and promoted release of the cathepsins in lysosome of shrimp [48]. However, it was evident that pressure caused protein denaturation and hence the cathepsin (B, D, H, and L) activities were inhibited. Finally, the protein denaturation and texture deterioration of shrimp induced by endogenous enzymes were significantly retarded by treatment with UHP above 300 MPa during storage.

  1. Section 3.4 . Changes in TVB-N and pH

We readjusted and supplemented the description of the relationship between cathepsin, protein decomposition and the shelf life of shrimp, and found that UHP treatment above 300 MPa for 5 min prolonged the shelf life of L. Vannamei by at least 8 days (Page 15, Line 431-435). We cited the study of Kaur et al, who found that the shelf life of higher pressure-treated black tiger shrimp was extended by 10 days during storage at 2°C (Page 15, Line 435-440). The descriptions in the manuscript are as follows: “The data showed that the activities of cathepsin (B, D, H, and L) and accumulation of the protein decomposition products during storage were significantly inhibited by UHP. It was worth noting that the shelf life of L. Vannamei was prolonged by at least 8 days after being treated with pressure above 300 MPa for 5 min (Figure. 5B). Kaur et al. [51] also found that treatment with pressure at 435 MPa for 5 min had the highest impact in reducing TVB-N produced in muscle tissue of black tiger shrimp compared with pressure at 100 and 270 MPa for 5 min. In addition, the study showed that the shelf life of shrimp treated at 435 MPa for 5 min was prolongedby 10 days during storage at 2 ± 0.5°C compared with that of the untreated samples.

Comment 2: Conclusions

The authors, in response to comment 4 on the conclusions section, state that they have added information in the results and discussion section. This is an inappropriate approach. The reader, after reading only the conclusions, should know what are the most important achievements of the published research without familiarizing himself with the entire article. Therefore, the conclusions section needs to be supplemented with at least two more research conclusions.

(Comment 2: Conclusions

The authors in this chapter summarized the effects of high pressure on shrimp. However, they only made one major conclusion. Authors should provide more conclusions, especially when they have done so much analysis.)

Response: Thanks for the reviewer's careful work. In section 4. Conclusions, we supplemented the description of the effects of different UHP treatments on protein degradation and oxidation, color and texture values of shrimp (1-2). We carefully revised the description of the conclusions that UHP treatment delayed the deterioration of protein properties and texture characteristics by inhibiting the activities of cathepsins, ultimately prolonged the shelf life of shrimp (3-4). Besides, we supplemented the description of the value and significance of the results in this study (5) (Page 15-16, Line 445-458). The description of the conclusions in the manuscript are as follows:

  • UHP treatment within 300 MPa for 5 min can maintain the natural qualities of shrimp, such as color and texture, and thus relatively prolong their freshness.”
  • “ UHP treatment above 300 MPa changed the natural protein structure of shrimp and caused protein degradation and oxidation, resulting in significant differences in its quality characteristics compared with fresh shrimp.”
  • “During storage, UHP treatment increased the permeability of mitochondrial membrane whereas decreased the stability of lysosomal membrane of shrimp. Subsequently, cathepsins B, D, H, and L were released but the enzyme activities were significantly inhibited by pressure above 300 MPa. Hence, the deterioration of protein properties and texture characteristics induced by cathepsins in shrimp were deferred during storage.”
  • “Generally, UHP treatment with 300 MPa for 5 min effectively maintained the stability of protein properties and quality characteristics, and prolonged the shelf life of stored Vannamei by 8 days.”
  • “ The results of this study might be helpful to understand the effects of UHP treatment on the endogenous enzymes, protein properties and quality characteristics of Vannamei during iced storage.”

Round 2

Reviewer 3 Report (Previous Reviewer 3)

Thank you to the authors for responding to the last two comments of the last review regarding the discussion of the results with the literature in the topic undertaken and the conclusions of the research. I believe that the authors have significantly and completely completed the missing sections of the manuscript. I have no further comments.

This manuscript is a resubmission of an earlier submission. The following is a list of the peer review reports and author responses from that submission.

Round 1

Reviewer 1 Report

The authors present the Effect of UHP and Storage Temperature on Endogenous Enzyme Activities, Protein Properties, and Quality Characteristics of Shrimp (Litopenaeus Vannamei). There are several research articles regarding the high-pressure effect on shrimp physicochemical quality, however, the effect on enzymes has not been studied comprehensively (that is a plus point). 

Please take appropriate action regarding the following comments;

Line 13: Mention enzyme name

Line 15: What is "W"? write full name first then use its abbreviation.

Line 16: Use a comma before "and" (same for Line 18, 20, 21, 46).

Line 32: Please specify the endogenous enzymes that are resulting in a rapid decline in freshness.

Line 33: Frozen storage for how much time?

Line 36: Drop loss or drip loss? and how protein denaturation caused taste decline? 

Line 41: Heating at what temp.?

Line 43: Foodborne enzymes?

Line 45: How much pressure?

Line 46: Replace "affects" with "causes"

Line 47: Could you explain "foodborne enzymes? As per my knowledge, there is no term like "foodborne enzymes". Please verify and use proper scientific terminologies throughout the manuscript.  

Line 47: Could you explain what type of "aquatic products?"

Line 48: Could you explain what type of "quality characteristics" are being greatly affected 48 by pressure treatment.

Line 50: Italicize the species name. you can also write as " L. Vannamei"

Line 54: for MFI, TCA, use full names first then abbreviated. 

Line 56: Storage for how many days?

Line 144: which enzyme?

Line 191: Replace "maybe" with "might be"

Line 192: Mention the high-pressure value and processing time. 

Line 201: "notably promoted" in a positive or negative way?

Line 202: Replace "maybe" with "might be"

Line 209: UHP was 209 more conducive to storage compared to which technology? 

Line 289: Replace "was" with "were"

Line 327: Replace "maybe" with "might be"

Line 345 & 365: How many days?

Reviewer 2 Report

This manuscript fits by the topic to journal Molecules produced by MDPI. Manuscript is well organized and almost all methods are well described.  I found only minor changes given in the following list.

-        - Line 39 There is missing declaration that UHP provide food sterilization and there are cited two papers that do not represent this technology. Reformulate this sentence in the sense that UHP is able to kill all living bacteria and input there reference on basic book describing UHP technology such as Hendrickx M. E. G and Knorr D. editors, Ultra High Pressure Treatments of Foods, Kluwer Academic/Plenum Publishers, New York, Boston, Dordrecht, London, Moscow, ISBN 0-306-47278-3, 2001st edition.

-       -  Lines 68-71 there is missing data about UHP machinery: volume of the chamber, temperature control system that is able to compensate temperature changes due to compression and decompression regime, time to reach given pressure etc. UHP parameters are not described in Table S1.

-        - Line 185 pressure within 200 MPa had not a moderate but “great” effect on texture!

-        - Table 1 page 6 Parameter W had received for 200 MPa extraordinaire value 550.05! Please, correct this value.

-     -  Table 1 parameter W reached for 500 MPa value for 9 days extremal value 68.44. Why? There is no comment!

-      -  Figure 1 C has no parameter symbol on scale y! Input there TCA and units description shift into the figure title.

-      -  Line 327 declares some theoretical assumption “It may be due to the protein was continuously decomposed by microorganisms…”. After UHP no living microorganisms can be found in the samples.

-     -  Line 347 Change there “PH” by “pH”.

Reviewer 3 Report

Introduction

Line 30-31- sentence needs citations

In this chapter, the authors should present current knowledge about influence off UHP on sea food, and highlight what new knowledge they want to gain and what problem to solve. The current state of knowledge and outline of the problem is not sufficient.

Materials and methods

2.1. Did the authors only run the UHP experiment in one repetition?

2.8. This subsection should be expanded.

Results and Discussion

The authors have made a lot of analyzes that I consider necessary and appropriate. The presentation of the test results is very good, it perfectly illustrates the qualitative changes of shrimps during storage after high pressure processing.

The results have been widely described, but there is no in-depth discussion, especially based on the literature on the effects of high pressure on seafood.

Conclusions

Line 350-351 – this sentence is repetition of information, please delete it

The authors in this chapter summarized the effects of high pressure on shrimp. However, they only made one major conclusion. Authors should provide more conclusions, especially when they have done so much analysis.

Round 2

Reviewer 3 Report

The authors addressed all comments but not all corrections were made satisfactorily. Some of the amendments need to be improved. In addition, the language of the corrections that were made needs to be checked by a native speaker - they contain many language errors. Below it states what still needs improvement.

2.1. Did the authors only run the UHP experiment in one repetition? - The authors wrote that they performed three repetitions of high-pressure processing. Please write this in the article in section 2.1.

Results and Discussion: The results have been widely described, but there is no in-depth discussion, especially based on the literature on the effects of high pressure on seafood.- What I meant was a deeper discussion of the authors' results with those published in the literature. The authors should expand this discussion as it is currently very poor.

Author Response

A letter outlining the changes with respect to the referees' comments

We would like to show our heartfelt thanks to the reviewers for their kind and helpful comments. According with these advice, we amended the relevant part in manuscript, and someone of the questions were answered below.

To reviewer 3:

Comment 1: Introduction

Line 30-31- sentence needs citations. In this chapter, the authors should present current knowledge about influence off UHP on sea food, and highlight what new knowledge they want to gain and what problem to solve. The current state of knowledge and outline of the problem is not sufficient.

Response: Thanks for the reviewer's careful work. We have carefully revised the expression: "Studies have shown that Litopenaeus Vannamei is rich in water and protein; in addition, shrimp is rich in protease, mainly distributed in the digestive system and muscle tissue." to " Studies have shown that Litopenaeus Vannamei is rich in water and protein, which provides favorable conditions for microbial growth [2]." (Page 1, Line 30-31), and "Ultra-high pressure (UHP) technology changes food composition and physicochemical properties with a pressure of 100-1000 MPa to achieve sterilization. [4, 5]. Previous studies found that raw shrimp would turn black due to polyphenol oxidase action during long-term storage, and the protein denaturation and cooking loss would happen during heating [6, 7]. As a non-thermal processing technology, UHP could maintain food quality and storage stability by inactivating foodborne enzymes and microorganisms [8]." to "Previous studies found that raw shrimp would turn black due to polyphenol oxidase action during long-term storage, and the protein denaturation and cooking loss would happen during heating at 100°C [5, 6]. Ultra-high pressure (UHP), as a non-thermal processing technology, is able to inactivate the live endogenous enzymes and microorganisms in aquatic products under the pressure of 100-1000 MPa, which effectively delaying the spoilage of aquatic products and finally prolonging their shelf life [7]. Moreover, UHP maintains food quality and storage stability by changing its composition and physicochemical properties [8]." (Page 1, Line 39-46) , and "Numerous reports have evaluated the effect of UHP on the basic physicochemical quality of aquatic products [9-11]. Moreover, pressure has been shown to significantly affects protein degradation, crosslinking and oxidation of muscle [12, 13]." to " At present, numerous reports have evaluated the effects of UHP on the basic physicochemical quality (such as moisture, texture, color, volatile substances, eta.) of aquatic products [9-11]. Additionally, the effects of UHP on the microorganisms and shelf life of aquatic products have also been reported [12, 13]. Moreover, pressure above 100 MPa has been shown to significantly causes protein degradation, crosslink-ing, and oxidation of muscle [1, 14]. " (Page 2, Line 47-52)

Comment 2: Materials and methods

2.1. Did the authors only run the UHP experiment in one repetition?

Response: We repeated the UHP experiment at least three times.

2.8. This subsection should be expanded.

Response: Thanks for the reviewer's careful work. We have carefully revised the expression: "The free sulfhydryl and disulfide bond contents of MP were evaluated as previously described [1]. The contents were expressed as nmol/g protein." to "The free sulfhydryl and disulfide bond contents of MP were evaluated as previously described with minor adjustments [1]. Brifly, 0.1 mL 5 mg/mL MP solution was mixed with 0.9 mL PBSF (3 mM EDTA, 1% SDS, 0.2 M Tris-HCl, pH 8.0) and 0.1 mL 10 mM DTNB, then the mixture was incubated at 4℃ for 1 h to detect free sulfhydryl content. Another 0.1 mL MP solution was mixed with 1 mL PBSd (8 M urea, 3 mM EDTA, 1% SDS, 0.1 M Na2SO3, 1% NTSB, 0.2 M Tris-HCl, pH 9.5) and incubated in the dark at 25℃ for 25 min to detect disulfide bond content. The absorbance of mixture was measured at 412 nm and the contents were expressed as nmol/g protein." (Page 3, Line 126-133)

Comment 3: Results and Discussion

The results have been widely described, but there is no in-depth discussion, especially based on the literature on the effects of high pressure on seafood.

Response: Thanks for the reviewer's careful work. We have supplemented the effects of high pressure on seafood in the introduction. (Page 2, Line 47-52)

Comment 4: Conclusions

Line 350-351 – this sentence is repetition of information, please delete it

Response: Thanks for the reviewer's careful work. We have deleted the sentence.

The authors in this chapter summarized the effects of high pressure on shrimp. However, they only made one major conclusion. Authors should provide more conclusions, especially when they have done so much analysis.

Response: Thanks for the reviewer's careful work. We have carefully revised the expression: "The data revealed that although the protein structural integrity was destroyed and protein oxidation was accelerated by UHP, the protein properties deterioration was effectively delayed during storage." to "The data revealed that the protein structural integrity was disrupted (MFI was increased), the exposure of internal hydrophobic sites and conformational changes of protein was promoted (surface hydrophobicity was increased), and the protein degradation was accelerated (TCA-soluble peptide was increased) by UHP. Moreover, the free sulfhydryl accumulation, the carbonyl groups formation, as well as the oxidation and complexation of tyrosine residues in protein was triggered by UHP. However, during storage, the protein properties deterioration was effectively delayed by UHP." (Page 14-15, Line 376-382). In addition, we have added some new descriptions of the relationship between protein oxidation and quality properties: "Due to the specificity of protein function, oxidative modifications can directly cause a variety of changes in food quality properties, such as hydrophobicity, solubility, water holding capacity, texture and digestibility [35-37]. Usually, the backbone or side chains of some amino acids (such as Arg, Lys, His, Asp and Pro) in proteins are oxidized directly or indirectly, known as protein carbonylation, ultimately leading to a loss of solubility and protein aggregation [38, 39]. In addition, aromatic amino acids like Tyr, Trp and His are very susceptible to oxidation. For example, dityrosine is produced from tyrosine and their metabolites [40]. The data indicated that protein oxida-tion of shrimp was significantly delayed by UHP during storage, resulting in the protein degradation and texture deterioration were effectively alleviated." (Page 9, Line 284-293), and "In addition, the protein oxidation in shrimp during storage was significantly delayed by UHP, which might be another reason why the quality deterioration of muscle was effectively suppressed." (Page 15, Line 388-390)
